# Virtual and Real-Time Synchronous Interaction for Playing Table Tennis with Holograms in Mixed Reality

**DOI:** 10.3390/s20174857

**Published:** 2020-08-27

**Authors:** Bin Wang, Ruiqi Zhang, Chong Xi, Jing Sun, Xiaochun Yang

**Affiliations:** Collage of Computer Science and Engineering, Northeastern University, Shenyang 110819, China; zhangruiqi1025@163.com (R.Z.); 15734022139@163.com (C.X.); sunjing@stumail.neu.edu.cn (J.S.); yangxc@mail.neu.edu.cn (X.Y.)

**Keywords:** mixed reality, holograms, human–computer interaction

## Abstract

Real-time and accurate interaction technology is required to realize new wearable Mixed Reality (MR) solutions. At present, the mainstream interaction method relies on gesture detection technology, which has two shortcomings: 1. the hand feature points may easily be obstructed by obstacles and cannot be detected and 2. the kinds of gesture that can be recognized are limited. Hence, it cannot support complex interactions well. Moreover, the traditional collision detection algorithm has difficulty detecting the collision between real and virtual objects under motion. Because location information of real objects needs updating in real time, it is easy to lose collision detection under high speeds. In the implementation of our system, Mixed Reality Table Tennis System, we propose novel methods which overcome these shortcomings. Instead of using gesture detection technology, we use a locator as the main input device and build a data exchange channel for the devices, so that the system can update the motion state of the racket in real time. Besides, we adjust the thickness of the collider dynamically to solve the collision detection problem and calculate rebound results responding to the motion state of the racket and the ball. Experimental results show that our method avoids losing collision detection and improves the authenticity of simulation. It keeps good interaction in real time.

## 1. Introduction

Mixed Reality (MR) technology, originating from Virtual Reality technology, is one of the hottest technologies in recent years. In the 1960s, Ivan Sutheland developed the earliest augmented reality system model (See-Through Head-Mounted Dispaly), which was the world’s first use of a CRT optical lens head mounted display [1,2,3,4,5]. But it lacked movement tracking [2]. Early in the 1990s, Tom Caudell and David Mizell formally put forward the terminology of Augmented Reality [1,3]. In 1997, Ronald Azuma proposed a broad definition of Augmented Reality technology [3].

MR devices, represented by Microsoft Hololens [6], have been developing in recent years. Even though the products of MR are gradually perfected, some shortages still exist. For example, Microsoft Hololens [7], as well as a representative work in an augmented reality device [8], can identify some user gestures and allow users to use one hand interaction. On the contrary, its shortcomings are obvious. It recognizes the characteristics of gestures, so the types of gestures that can be recognized are limited. The device can track the movements of the hand, but cannot get the coordinates of the hand accurately [9]. The choice of virtual targets depends on helmet movement, which restricts the implementation of complex interaction.

Large companies have devoted to enhance human–computer interaction, especially in motion capture. Microsoft Xbox uses a somatosensory camera (Kinect) to detect the movements of the user’s limbs and makes games such as fruit cutting, however it cannot capture the changes of the user’s finger accurately [2]. Though the device can roughly capture the change of the position of the hand, it is difficult to capture the rotation. Oculus launched Finexus which could liberate both hands, but the identification distance was short and there was a lack of mature equipment [2]. Judith Amores et al. [10] implemented painting in MR by utilizing Hololens to pinpoint the position of the finger. Christina Pollalis et al. [11] presented HoloMuse to enjoy museum collections through gesture-based interaction with Holograms, which provided the interaction of moving and rotating exhibits. These systems can accomplish some interactive task but the interaction is limited to several gestures. It can only achieve single target selection and slow drag functions. J.M.Huang et al. [12] used a wireless mouse and pressure sensor as input device, and designed a new interactive algorithm of multi-object selection, model cutting and component mobile, to achieve Finite Elements Analysis in MR. Cedric Kervegant et al. [13] made use of haptic devices to provide user tactile experience in Mid-Air.

The real-time requirement of MR technology are not only high in the field of games, but also in medical experiments [14,15], aerospace [16], production engineering [17] and other fields. For example, medical experiments [15] simulated through VR environment have a very high requirement on real-time application. If there is a little delay, it will lead to the failure of experiments. Delays in response to operations in the space sector can also have serious consequences [16]. So our approach is not limited to games, but is suitable for all other areas with high real-time requirements. Some existing interaction technologies are known to have latency problems, and the latency problem itself is the limitation of hardware. At the same time, the authenticity of the existing MR Products is often not fully considered, but in our system, both real-time performance and some necessary factors of the real scene are considered, such as the rotation and hitting of the ball.

VR System for Neurorehabilitation [14] only considers virtual technology and does not combine real technology, so the real experience of the scene is poor. Meanwhile, because it is mainly targeted at Neurorehabilitation experiment, it does not consider the real-time problem in the experiment. MR For Feline Abdominal Palpation Training is an early application of mixed reality technology. Due to the limitation of computing ability of early devices, many necessary experimental operations are ignored in [15]. The works in [16,17] mainly emphasize the application of Augmented reality in the field of automobile and aerospace industries. It mainly involves some basic experiences, but does not involve high-real-time operations, such as parts repair and virtual driving. At present, most of the papers have avoided the real-time problem, because it is limited by the hardware conditions, and the real-time factor ignored by many systems will inevitably lead to poor experience (reduced authenticity). However, we solve the device delay problem through the corresponding algorithm, to bring better experience for users.

In order to meet the user’s requirements for better interaction, we provide a good interactive system. By wearing MR equipment (Microsoft Hololens) and holding a table tennis bat, the user can experience the interaction with the virtual ball. A series of interactions simulate the real batting action, including juggling, hitting a ball against a wall and other conventional actions. This system realizes real-time, accurate interaction and eliminates several tedious acts such as picking up the ball. What is more, we assume that the virtual ball will not be disturbed by the wind in the real environment.

There are several difficulties in the whole work:

1. The MR device lacks a matched locator. In order to make the imaging device work with the locator, we need to provide useful information collected by the locator to the imaging device by the computer.

2. It is hard to detect the collision between the ball and the racket when the racket is moving fast. As we known, the location information collected by the locator is transmitted to the computer frame by frame, when the racket moves fast in a short period of time, the position differences between each sample point may be very large, therefore it is difficult to accurately locate the position of the fast-moving racket and interact in real-time.

3. In order to realize the real physical interaction, users should be able to control the racket and hitting power to adjust the angle and velocity of the ball flight. However, the computer only obtains the discrete data of the racket position and we cannot simply modify the material of the collider to achieve the normal rebound effect. Therefore, we need to use these discrete data to simulate continuous motion and calculate the rebound direction and velocity of the ball after each shot.

4. It is also difficult to calculate the speed and direction of the spinning ball. According to the rotation of the ball, the different positions of the ball and the direction of movement of the ball hitting the racket should be considered.

## 2. System Design

The method introduced in this paper aims at realizing the simulationinteraction between a real table tennis racket and virtual table tennis in a mixed reality environment. It consists of four steps: 1. Data acquisition: through HTC Vive Controller to locate the racket, extract the information of Transform components, complete data acquisition; 2. Data Exchange: establish data exchange channel and connect HTC Vive and Microsoft Hololens through Unity3D engine. The purpose is to read and write related parameters in real time; 3. Execute collision detection: determine the position relationship between the racket and the virtual ball in real time, and return the collision result; 4. Output physical rebound: when the collision detection algorithm returns to the collision, the physical rebound algorithm is executed to calculate the rebound speed of the virtual sphere and output to the Hololens display screen. Meanwhile, we also consider the rotation of the ball.

### 2.1. Selection of Input Devices

In order to meet the needs of users to play table tennis in MR environment, we need to use appropriate motion capture or spatial positioning technology. At present, although the infrared tracking technology of Microsoft Hololens can capture the position of the hand, the hand is easily blocked by the racket in the actual operation process, resulting in tracking failure. In addition, it is difficult for Hololens to capture the rotation of the hand, so it is difficult to detect the rotation of the racket. Considering these, we need to use other tracking and positioning devices. Up to now, the working principle of the mainstream VR equipment mainly relies on the following spatial positioning or motion capture technologies [18,19,20,21,22]: Laser locating technology, Infrared optical locating technology [23,24,25], Visible light locating technology, Ultrasonic locating technology, Computer vision motion capture technology and Motion capture technology based on inertial sensors [2,26,27,28]. A comprehensive comparison of these technologies is shown in Table 1. These performance comparisons are mainly derived from the analysis of the product description [29,30,31], and we use HTC Vive [29] to track the racket. It has the advantages of high locating accuracy, unaffected by occlusion, fast response speed, supporting multiple target locating and wide coverage [8]. Compared to other commonly used VR devices, such as Oculus Rift, PS VR, etc., it supports wider coverage and gives players more room to move. Meanwhile, it is more accurate than PS VR, which is very important in the player experience. It is also the most practical device currently applied to VR.

Now, the system can capture the movement of the racket in the space in real time. However, it is not enough to realize the interaction of the batting action. First, the imaging device we need to use is incompatible with the locating device. Second, when the user is quickly waving a racket to hit the ball, the virtual ball often passes through the racket to the ground. The faster the racket swings, the more prone to error it is. Third, when the user is juggling, the user cannot control the racket and the hitting power to adjust the angle and velocity of the ball flight. To put it simply, the racket has not exerted force on the ball. How to solve these problems is the focus of this section.

### 2.2. System Architecture and Device Deployment

The whole interactive system uses HTC Vive controller as the main input device, and we do not need to use buttons, we just use it as a locator. If there are smaller locators in the future, we can replace it directly. The controller collects the square data of the racket frame by frame and provides it to Unity3d engine in the form of Transform component data. We only need the Position and Rotation parameters. Hololens is not only an imaging device, but also an auxiliary input device. It is used to start operation, pick up balls, move the view angle and scan terrain. Due to the limitation of hardware compatibility, they cannot run under the same project, but we can still use them at the same time.

As shown in Figure 1, the relationship between users and devices is established. First, we use a workstation to run two Unity3d projects simultaneously, supporting two sets of peripherals. One project connects HTC Vive Controller, which receives the Transform information collected by the locator, and the other project connects Microsoft Hololens and calculates the results of collision detection and rebound. A text document can be used to transfer Position and Rotation parameters in real time between the two projects. Unity3d engine can read and write parameters in real time through edited C# script. Next, we use Holographic Remoting Player [32], a Microsoft Hololens suite to render Unity3d images in real time on Microsoft Hololens. Wi-Fi is used to connect Holographic Remoting Player and Microsoft Hololens. Microsoft Hololens can also send part of the input information to the workstation, such as user movement, view change, terrain of real environment and user gestures, voice input. Finally, the mixed reality image is passed to Microsoft Hololens. In this way, we have successfully combined them.

### 2.3. Coordinates Unification

The interactive system uses two Unity3D projects to support HTC Vive and Hololens devices respectively. The two projects correspond to two 3D scenes, which are relatively independent. Therefore, external files are needed to support coordinate data transmission. Obviously, each scene has its own coordinates. How to unify the two coordinates so that the position and rotation of the real racket can be directly used in the MR is explored in this section. As we know from experiments, the spatial coordinates of MR are determined by the position and rotation of the Hololens in the initial stage of operation. Therefore, the initial state of each interaction is different and the world coordinates are different. Assuming that the position of the racket (locator) is $p_{v}$, the rotation $r_{v}$, the corresponding collider position $p_{m}$ and the rotation angle is $r_{m}$ in the VR, there exists transfer variable *T* to satisfy:(1)$$\left[p_{v},r_{v}\right]*T=\left[p_{m},r_{m}\right]\;$$

Before interaction, *T* needs to be obtained to realize coordinate unification and ensure the normal operation of the interaction. In summary, a reference model is set up in front of the user’s perspective with position $p_{m}$ and rotation $r_{m}$. When the user runs the interactive system, he/she needs to move the real racket to make it consistent with the reference model, and click confirm. The system records the Transform data of the locator at the moment. Transfer variable *T* can be calculated according to Equation (Equation 1). Now, the two projects have completed coordinates unification. In the subsequent interaction, the interactive system can keep the collider of the racket consistent with the real racket at all times, which provides the premise for collision detection.

## 3. Algorithms

In an MR system, in order to ensure the authenticity of the virtual world, collision detection must have high real-time applicability and accuracy. For the so-called real-time feature, based on the requirements of visual display, the frequency of collision detection can reach at least 24 Hz, and based on the tactile requirements, the speed of collision detection can reach at least 300 Hz to maintain the stability of the tactile interaction system, and only to achieve 1000 Hz to achieve smooth effect [27]. The requirement of accuracy depends on the requirements of a virtual reality system in practical applications. In the mixed reality, hitting the table tennis ball is a simple action which cannot be separated from the collision detection algorithm. After calculation, according to the traditional collision detection algorithm, the collision detection frequency needs to reach more than 100 Hz. Only in this way can it simulate the actual effect of playing a ball more stably and avoid errors in interaction. However, it is far from reach for most personal computers, especially Microsoft Hololens.

### 3.1. Collision Detection

#### 3.1.1. Analysis of the Cause of Losing Collision Detection

A game or interaction system, which involves the interaction of virtual objects, is inseparable from collision detection. In game development, collision detection methods are often used by game engines, and parameters are adjusted according to needs. It can solve the collision problem between most virtual objects, but no matter which type of detection is used, there will be corresponding defects.

We only consider hitting the ball with one side of the racket (not chopping, which means the normal vector of the racket surface is close to the hitting direction). The error of the virtual ball passing through the racket always occurs when the racket is swung fast. This error is more likely to occur when the frame rate is low. Because the locator does not transmit the location information continuously, but only refreshes the location information each frame, and there is a time interval between the frame and the next frame. When the racket is moving quickly, the location information of the two frames refreshed before may be quite different. However, the volume of the table tennis ball is relatively small. Take FPS = 50 and the swing velocity = 10 m/s as an example. The system captures position information every 0.02 s, the average interval of each collection is 20 cm, which is about 5 times the diameter of the virtual ball (standard table tennis diameter is 4.4 cm). It is possible for the ball to pass through the racket because of the failure to detect the collision. As shown in Figure 2, the position information received from the locator happens to be missed by the ball in 5 consecutive frames.

#### 3.1.2. Trajectory Error Analysis and Trajectory Simulation

In the interactive process, the racket is tracked in the form of several sampling points. Figure 3a shows the racket state in each frame while swinging. They can be viewed as a set of points in space, each point contains position data p and rotation data r as shown in Figure 3b. According to these points, we can roughly imagine the trajectory of the racket. The black curve in Figure 4 represents the real trajectory of racket in a swing action, the red points represent the collected samples. The sampling interval between the two sample points is not more than 0.1 s (determined by the frame rate). In such a short period of time, the direction of motion will not change considerably. So between the sampling points, we can see it as a linear motion and simulate a new trajectory (the blue lines). According to the simulated trajectory, we can calculate a time interval of the swing angle according to the position information of the adjacent sample points, and calculate swing velocity combined with the frame rate. In order to deal with the collision problem easily, we can directly increase the thickness of the collider to determine whether the racket is colliding with the ball, which is the most efficient method to ensure the real-time performance of the interaction.

#### 3.1.3. Specific Process of Collision Detection

First, bind the locator to the racket. We set a collider at the position of the racket in the 3D scene, it needs to match the size of the racket surface. The collider is used to detect whether the racket is colliding with the virtual ball. The locator will detect the movement of the racket and adjust the posture of the collider in real time as shown in Figure 5. Thus, the user can slowly move or rotate the racket in his hand and let the virtual ball roll on the racket. In this process, the system records the position of two frames of racket $p_{b}$, $p_{l}$ at any moment. The distance between $p_{b}$ and $p_{l}$ is expressed in *S* and the the velocity of the racket is expressed in *v*. In the actual situation, they satisfy
(2)$$S=\int_{t_{2}}^{t_{1}}\;dx\;$$

However, the velocity of the racket at each time is unknown. In order to better regional relations between symbols, you can refer to Figure 6. Based on the simulate trajectory, we can calculate the average velocity between any two frames according to the formula:(3)$$\left|p_{l}-p_{b}\right|\approx \int_{t_{2}}^{t_{1}}\;dx\;$$
(4)$$\bar{v}=\left|p_{l}-p_{b}\right|/\delta t\;$$
where δt is time interval between two frames, $p_{b}$, $p_{l}$ are three-dimensional space coordinates, $p_{l}-p_{b}$ is a three-dimensional vector. The value of *v* can be expressed as
(5)$$v\approx \bar{v}=\frac{\sqrt{{\left(x_{l}-x_{b}\right)}^{2}+{\left(y_{l}-y_{b}\right)}^{2}+{\left(z_{l}-z_{b}\right)}^{2}}}{\delta t}\;$$

Then, we need to adjust the thickness of the collider in real-time according to the velocity of racket. The larger the $\bar{v}$ (the faster the racket moves), the greater the thickness of the collider. Figure 7 shows the collider of the size of different swing velocity (green area, not visible in actual interaction). In order to ensure normal interaction, any point on the track needs to be detected by the collider, and the thickness of the collider th should meet:(6)$$th=\left\{\begin{array}{cc} v*\delta t, & \mathrm{if}\;{\bar{v}}^{\prime }<threshold\\ (v+{\bar{v}}^{\prime })*\delta t & \mathrm{if}\;{\bar{v}}^{\prime }\geq threshold \end{array}\right.\;$$${\bar{v}}^{\prime }$ is the average velocity of ball. At any time, the virtual ball is also moving, so we have to consider the error of the ball’s displacement to the collision. However, when the ball speed is relatively slow, the influence of the ball speed can be ignored.

However, simply increasing the thickness of the collider will also bring some errors. As shown in Figure 8, when the racket moves fast, the thickness of the collider corresponding to the racket will increase. At this time, the virtual ball just flies into the collider from the side of the collider, and the system will be directly misjudged as collision.

In view of this situation, we set an angle variable in the system to detect and record the angle between the virtual ball and the racket surface normal vector in real time. Assuming that the center position of the racket surface is $p_{0}=\left(x_{0},y_{0},z_{0}\right)$, the normal vector of the racket surface is $\vec{n}=\left(x^{\prime },y^{\prime },z^{\prime }\right)$, and the center position of the ball is $p_{1}=\left(x_{1},y_{1},z_{1}\right)$, there are:(7)$$\left(p_{1}-p_{0}\right)\cdot \vec{n}=\left|p_{1}-p_{0}\right|*\left|\vec{n}\right|*\cos\theta \;$$

From the position of the collider relative to the racket surface, it can be seen that when collision is detected, the virtual ball is located behind the racket surface, θ must be greater than 90 degrees. At this time, it is only needed to judge whether the ball in the previous frame is in front of the racket, that is, whether θ is less than 90 degrees. If it is less than 90 degrees, the result of collision should be executed. Otherwise, the result of collision should be neglected. Figure 9 shows the difference between the two cases. In practical interactions, just compare the value of $\left(p_{1}-p_{0}\right)\cdot \vec{n}$ and 0, which can check whether the collision result is effective. It greatly reduces the amount of calculation.

### 3.2. Simulate Ball Bounce

When hitting a ball in a real environment, the contact between the ball and the bat will last for a few milliseconds. During that time, the ball will deform, and the deformation of the ball will directly determine the trajectory of the ball after bouncing. It is in these few milliseconds that the players can control the movement of the racket to change the deformation of the ball and then adjust the trajectory of the ball. In MR, we cannot restore the deformation process of the ball. This requires complex calculations and the device refresh rate is not sufficient to collect relevant data. Through the above steps, we can avoid the virtual ball passing through the racket while swinging. But we also need to bounce the ball according to the user’s swing velocity. Simply replacing the ball’s physical material can only let the ball bounce according to their own velocity and the user cannot change the ball’s flight rate by his/her own behavior.

According to the previous section, the interactive system detects collisions frame by frame according to edited scripts. When the game engine detects a collision, the instantaneous velocity of the rebound ball ${\vec{v_{1}}}^{\prime }$ is satisfied:(8)$${\vec{v_{1}}}^{\prime }={\vec{v_{0}}}^{\prime }B+\vec{v}\;$$
where *B* is a rebound coefficient (Default takes 0.75) and $\vec{v}$ is the speed of the racket. The size of ${\vec{v_{0}}}^{\prime }$ is equal to the virtual ball’s velocity $v^{\prime }$:(9)$$v^{\prime }=\frac{\left|p_{l}^{\prime }-p_{b}^{\prime }\right|}{\delta t}\;$$

The direction is opposite to the normal coordinate of the beat surface. When calculating the direction, the transfer matrix *T* is calculated based on the beat surface normal vector $\vec{n}$:(10)$$\vec{n}\cdot T=\vec{z}\;$$
(11)$${\vec{v}}^{\prime }\cdot T={\vec{v}}^{\prime \prime }\;$$$\vec{z}$ is is the unit vector of z axis direction. Reverse the z axis coordinate value of ${\vec{v}}^{\prime \prime }$ and easily obtain ${\vec{v_{0}}}^{\prime \prime }$, which is the reflection of ${\vec{v}}^{\prime \prime }$. After inverse transformation:(12)$${\vec{v_{0}}}^{\prime }={\vec{v_{0}}}^{\prime \prime }\cdot T^{-1}\;$$

The whole process is shown in Figure 10. First, the racket’s local coordinates are rotated so that the normal vector $\vec{n}$ coincides with the z-axis direction. At this time, the ball’s flying speed is rotated as ${\vec{v}}^{\prime \prime }$. After reversing the z coordinate value, the rebounded speed ${\vec{v_{0}}}^{\prime \prime }$ is obtained, and then rotate to the original local coordinates to ${\vec{v_{0}}}^{\prime }$.

The relationship between speed is shown in Figure 11, hen the racket moves vertically upward at a certain time, the speed of the virtual ball makes an angle of 30 degrees to the racket surface. The instantaneous speed of the rebound is shown as ${\vec{v_{1}}}^{\prime }$. The whole calculation process only uses the information of the Transform component of racket and virtual ball at multiple time points. According to the above steps, users can easily control the batting intensity.

### 3.3. The Rotation of The Ball

In the motion of the ball, the movement of the ball itself has two forms of rotation and non-rotation. In Section 3.2, we only considered the case where the ball does not rotate, and the problem of the rotation of the ball is not involved. However, in order to make the system more in line with the actual situation, we also consider the rotation of the ball in the system design. The rotation of the ball includes top spin, the back spin, and side spin. The rotation of the ball obeys Bernoulli’s theorem and the parallelogram rule.

#### 3.3.1. The Creation of Spinning Ball

In Section 3.2, we only analyze the direction and speed of the ball rebound, but do not consider the rotation of the virtual ball when the ball collides with the racket. When the ball collides with the racket, the generated force can be decomposed into forces in two directions, one being the force $F^{\prime }$ perpendicular to the racket through the center of the ball, and the force *f* which is parallel to the racket, as shown in Figure 12. Since the speed and direction of the ball bounce has been obtained in Section 3.2, this part is only to solve the rotation of the ball. Rotation is only related to *f*, and has nothing to do with $F^{\prime }$, so we only need to consider the size of *f*. However, it is not easy to find force *f* in the system, so in order to simplify the calculation, the sub-speed in the horizontal direction of the racket (the speed of the racket motion can be decomposed into the direction perpendicular to the racket and parallel to the racket direction) replaces the *f*. The speed of the parallel racket direction is recorded as $v^{\prime }$, therefore, the rotation of the ball ω satisfies
(13)$$\omega =k^{\prime }\frac{v^{\prime }}{R}\;$$
where *R* is the radius of the ball, and $v^{\prime }$ is used to replace *f*, in order to enhance the reliability of the equation, the equation is multiplied by the coefficient $k^{\prime }$. Through the above formula, we can find the rotation of the ball after the racket collides with the ball.

#### 3.3.2. The State of Spinning Ball

Looking from the left side of the table to the right, counter clockwise rotation is called top spin, and the clockwise rotation is back spin; when looking down, clockwise rotation is a ball with left spin and counterclockwise rotation is a ball with right spin. For the analysis of the ball with top spin, in reality, the top spin rotates forward along the transverse axis when flying in the air, so the ball will drive the surrounding air to rotate with it. When the ball flies forward, the air circulation above the ball is opposite to the resistance of oncoming air, resulting in a slower air flow rate in this area. The air circulation under the ball is consistent with the resistance of oncoming air, and the air flow rate in this area is larger. According to Bernoulli’s theorem, the pressure is lower in places with large flow rates, and the pressure is higher in places with small flow rates. Therefore, the pressure on the top of the ball is high, and the pressure below is low, resulting in a downward force F. Since the front and rear circulation of the ball is perpendicular to the direction of the air flow, no pressure difference is generated. According to the parallelogram rule, the resultant force *F* can be obtained, as shown in Figure 13. In the system design, according to the rotational speed ω of the top spin, the pressure difference *F* and ω satisfy
(14)F=kvω
where *v* is the speed of the ball itself, it can be understood that the faster the ball is, the greater the effect with the air resistance, so the difference between the upper and lower resistance *F* is greater. In the meantime, the faster the rotation speed, the larger the *F* is, so multiply by a coefficient *k*. The gravity of ball $g^{\prime }$ is:(15)$$g^{\prime }=g+\frac{F}{m}\;$$
where *g* represents the gravitational acceleration of the Earth’s surface and *m* represents the weight of the ball. The principle of the ball with back spin is the same as that of the ball with top spin, and the distinction between them is $g^{\prime }$:(16)$$g^{\prime }=g-\frac{F}{m}\;$$ For the ball with top spin, the ball with back spin, and the normal ball, the main difference is the different downward force, resulting in different downward accelerations, as shown in Figure 14.

The principle of the side spinning ball is actually the same as the ball with top spin. For the left spinning ball, there is no change in the airflow between the upper and lower edges of the ball during flight, the front and back of the ball are perpendicular to the direction of air flow and there is no pressure difference, however, the left air circulation of the ball is different from the right air circulation, creating a rightward pressure difference, as show in Figure 15. At this time, the force *F* is satisfied (14), and a right acceleration a is generated, which is equal to the force *F* divided by the mass, and the principle of right spinning ball is the same as the left spinning ball.

## 4. Experiment

### 4.1. Methods

This section is the experimental part of this paper. The main input device used in the experiment is HTC Vive Controller. The MR output device is Microsoft Hololens developer version. It carries an Intel Atom x5-Z8100 processor and GPU for HoloLens customized by Microsoft. The game engine used is Unity3D 2017.1.1 version. The graphics workstation running the game engine is a DELL Precision Tower 5810 equipped with Windows 10 operating system and GTX 1080 GPU. This experiment will quantitatively test the FPS of the interactive system under different swing frequencies, sample the complete swing motion trajectory points, draw the simulation trajectory, and qualitatively test the delay and collision accuracy.

Frame rate (or FPS) is called the frequency or rate at which bitmap images of frames appear continuously on the display. The frame rate is closely related to the user’s immersive interactive experience. The temporal sensitivity and resolution of human vision vary according to the type and characteristics of visual stimuli, and vary among individuals. The human visual system can process 10 to 12 images per second and perceive them separately, while the higher rate is regarded as motion. For a game screen, especially first person games, more attention is paid to the level of FPS. If FPS is less than 30, the game screen will appear incoherent, and the interaction of mixed reality will pay more attention to FPS. The system introduced in this paper uses HTC Vive Controller to track and locate the table tennis racket. The collected data are transmitted to Unity3D engine frame by frame. The frame rate directly determines the frequency of data acquisition, and then affects the accuracy of collision detection and physical simulation. In order to prove that the proposed algorithm can provide better interaction fluency and user experience, this section will test the FPS results of two collision detection algorithms at different swing frequencies and make a comparative analysis.

### 4.2. Results

At the beginning of the experiment, the user needs to adjust the position of the racket to match the initial position of the marked collider, and then click ahead to start the interaction. Transform component information obtained by HTC Vive Controller will be transmitted to Unity3D through edited scripts, which will detect collisions frame by frame and calculate physical rebound based on the results. The user picks up the virtual ball by clicking on it. At this time, the ball will automatically move to the front of the user’s field of vision. Click again, the ball will be applied gravity. The whole process can be repeated.

Figure 16 is the experimental result of the frame rate test using collider dynamic adjustment algorithm. Microsoft Hololens display image frame rate changes at different swing frequencies can be observed. From top to bottom are: the racket is static, the FPS is 53.25, frame rate is stable; FPS is 55.11 after 60 swings; the FPS is 53.18 after 100 swings, frame rate fluctuation has increased slightly; the FPS is 48.49 after 140 swings, frame rate fluctuates slightly. Overall, the FPS that the user sees is always maintained at around 50. Even if the swing speed reaches 140 times per second, it can still maintain a frame rate above 45.

In the interactive process, Unity3D engine automatically records the position and rotation angle of the racket frame by frame. Each record is a Transform component information. In the experiment, the record collected each time is regarded as a sampling point in space, and projected to the 3D coordinate system according to the position parameters. A series of sampling points is formed. Finally, the adjacent sampling points are connected by line segments and the simulated trajectories are plotted. The integrity of simulated trajectories will directly affect the reliability of collision detection and physical effects simulation results. In the Unity3D, the table tennis racket moves along this path.

Figure 17 is a complete and fast swing trajectory simulation experiment result, including 12 sampling points, about 0.25 s. From the pictures of different angles in the figure, it can be seen that the simulation trajectory is close to a complete curve. It shows this system can basically restore the racket trajectory, which provides a guarantee for collision detection and Simulation of physical effects.

Figure 18 shows the result of the user’s control of the racket movement in the actual environment and the position of the racket detected by the computer. In the testing process, the interactive real-time performance is good and the user does not observe the delay. It proves that HTC Vive is a suitable input device and the connection method we use is effective and stable.

Figure 19 shows a complete interaction process. Because of the need to use the Microsoft Hololens built-in video recording system, in order to better display the experimental result, the experiment used a racket model to enhance the real racket. The experimental results show that the collision detection between racket and virtual ball is accurate and stable, and the problem of collision detection loss will not occur. After a short period of practice, the user can control the ball freely. The defect is that once the virtual ball flies out of the window, users cannot observe the movement of the virtual ball for the first time, which affects the judgment of catching the ball, because the window of Hololens is too small. This will affect the user’s interactive experience.

Figure 20 shows the track of the spinning ball after hitting the racket. Because it is difficult to directly capture the rotation of the real-time situation, a diagram is used to simulate the real movement and the rotation of the different ball with the racket after collision, rotating the ball with the racket to produce a corresponding friction, and the friction and the racket rebound will work together in the ball movement, resulting in the ball taking different trajectories.

## 5. Conclusions

The experimental results show that the proposed algorithm achieves high accuracy and real-time interactive table tennis activities in a hybrid reality environment. Compared with the actual environment, users can pick up the ball by clicking, and the virtual ball will not be disturbed by wind. Even if there is no way to use fully compatible input and output devices, through text reading and writing to transfer data, input will not be significantly delayed. Because each input and output device has its own local coordinates, the relationship between local coordinates and world coordinates depends on the specific position at the beginning of the interaction, so the initial position of each interaction is uncertain and the local coordinates are different. Users need to adjust manually according to the actual situation, so that the position of the racket can correspond to the actual situation and the colliders coincide. At present, due to the limitation of Microsoft Hololens window range, users can easily fly out of the window after hitting the ball, thus affecting the user’s observation of the movement of the ball, thereby affecting the judgment of catching the ball, and to a certain extent, affecting the user’s interactive experience. In addition, when deploying hardware, it should be noted that there will be some interference between the two sets of devices, and the location of the locator needs to be adjusted to achieve the best results before interaction. Briefly, interaction can be carried out stably in mixed reality environment.

This paper proposes algorithms for direct interaction between real and virtual objects in view of the limitations of hybrid reality interaction technology. At present, the interactive system only supports a single user, and the interactive algorithm only supports the ordinary hit action. In the future work, we will continue to explore this interactive technology to achieve more complex interactions, such as: the simulation of complex movements such as a ball with top spin, chopping the ball, and so on. The rotational speed of the virtual ball and the sliding friction between the racket and the ball are taken into account. In addition, Hololens has provided the function of sharing virtual scenes among multiple devices. Next, aiming at the input–output data synchronization algorithm, we will implement the multi-user interaction system with the scene in the mixed reality environment, which will bring the hybrid reality interaction technology to a higher level and bring convenience to the simulation of various human activities.

At the same time, this technology cannot only be used in this system, but also in the application environment that requires high real-time and accuracy, such as medical science, education and other fields. Therefore, we will spend more time and experience to expand the application scope of this technology in the future. At the same time, in order to make the user’s experience more real, the lighting of the scene, wind resistance and other factors encountered in reality will be considered in the follow-up work.

## Figures and Tables

**Figure 1 sensors-20-04857-f001:**
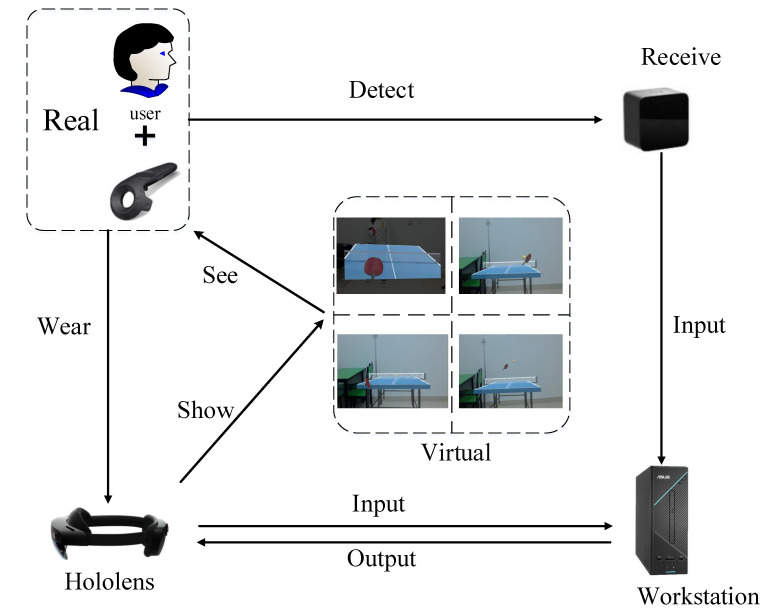
Interaction relationship between user and devices.

**Figure 2 sensors-20-04857-f002:**
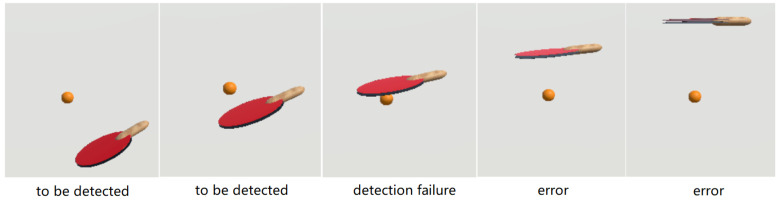
Interaction relationship between user and devices.

**Figure 3 sensors-20-04857-f003:**
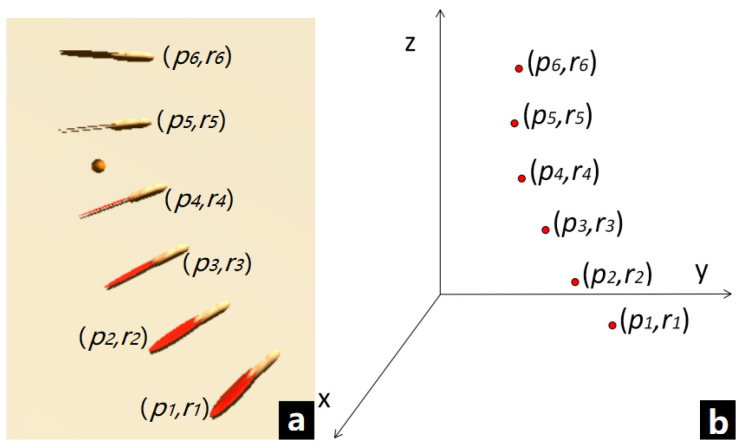
(**a**) The racket state in interactive process; (**b**) The sampling points distribution model.

**Figure 4 sensors-20-04857-f004:**
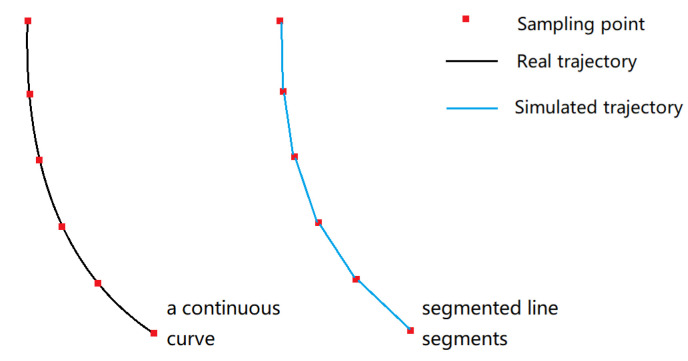
Real trajectory and simulated trajectory of fast swing.

**Figure 5 sensors-20-04857-f005:**
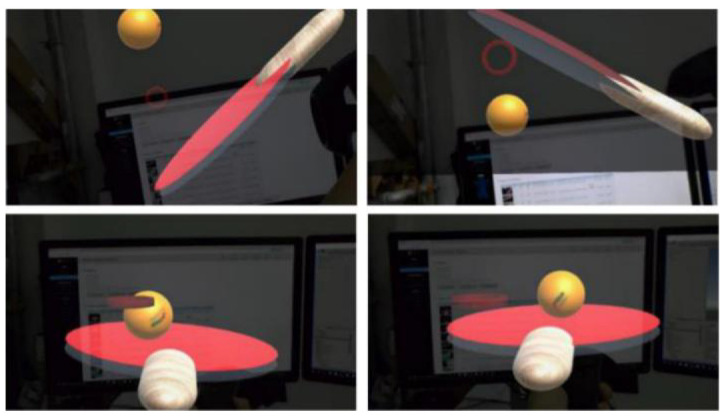
Four figures show the collider follows the action of the user in order to complete the contact interaction.

**Figure 6 sensors-20-04857-f006:**
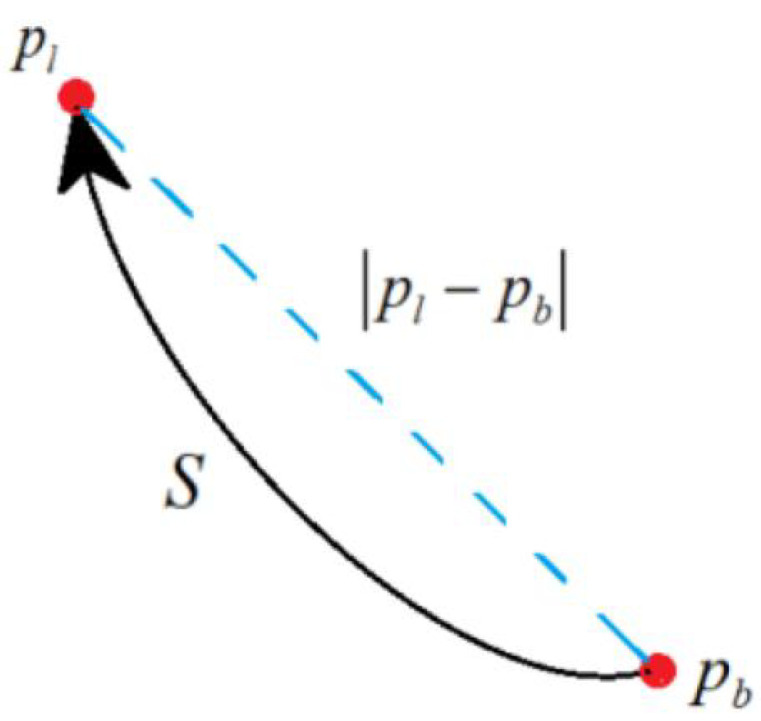
The relationship between symbols.

**Figure 7 sensors-20-04857-f007:**
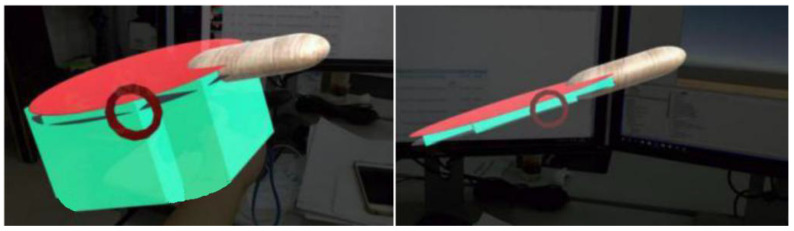
(**left**) Fast move thickness; (**right**) Slow move thickness.

**Figure 8 sensors-20-04857-f008:**
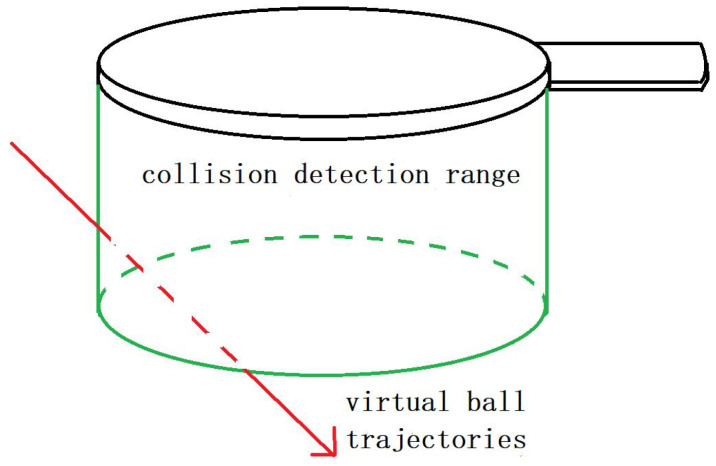
Collider error detection.

**Figure 9 sensors-20-04857-f009:**
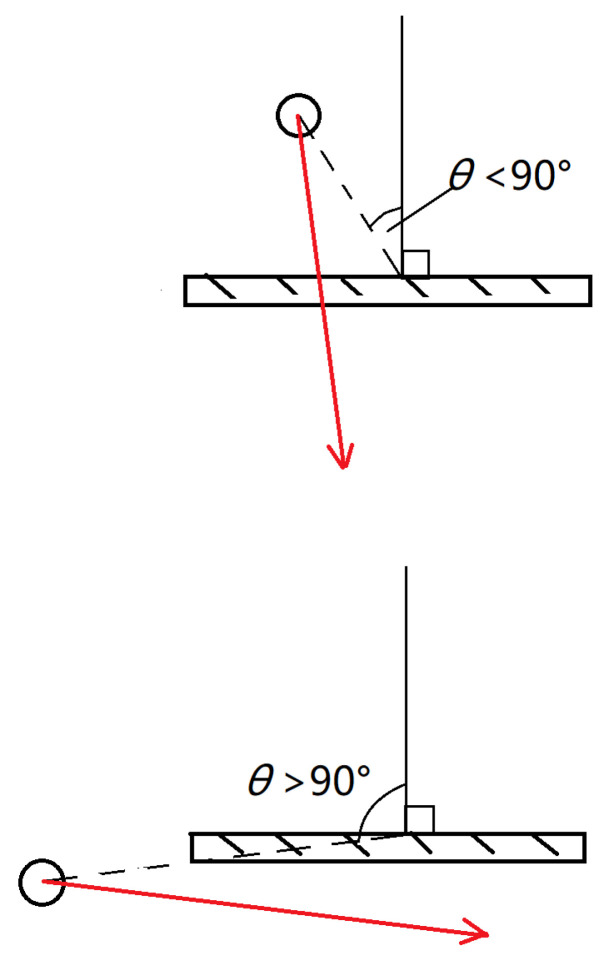
(**top**) Less than 90 degrees case; (**bottom**) Greater than 90 degrees case.

**Figure 10 sensors-20-04857-f010:**
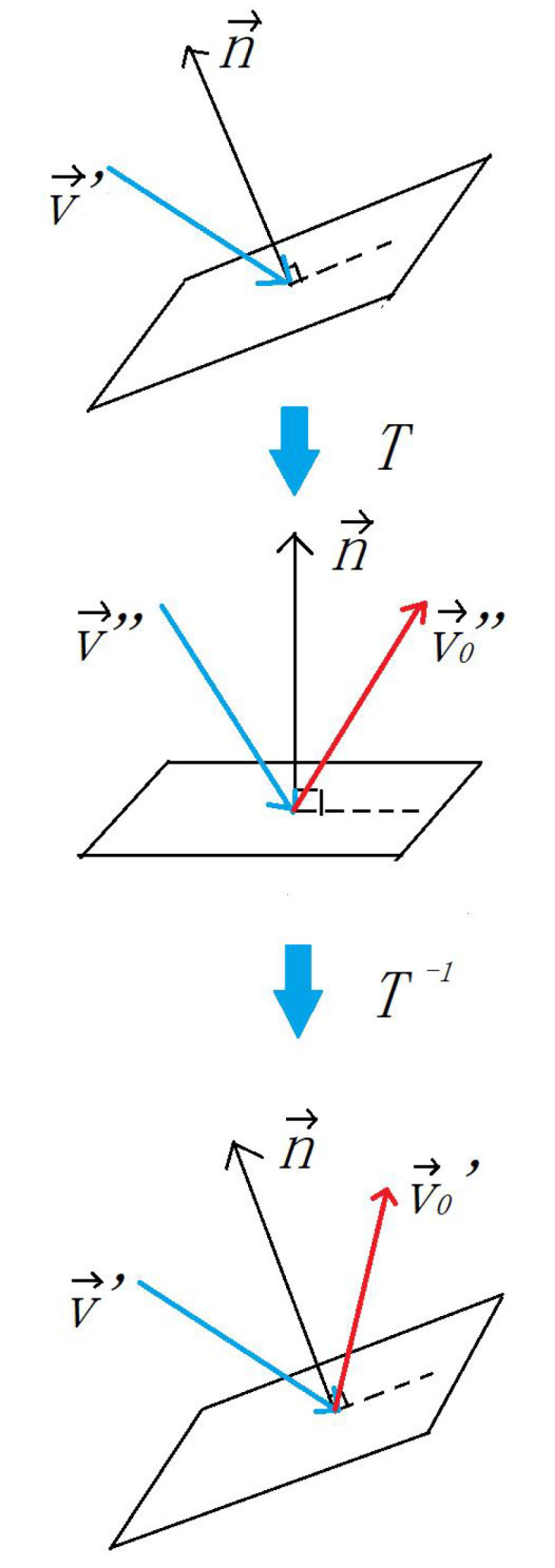
Calculating the direction of rebound.

**Figure 11 sensors-20-04857-f011:**
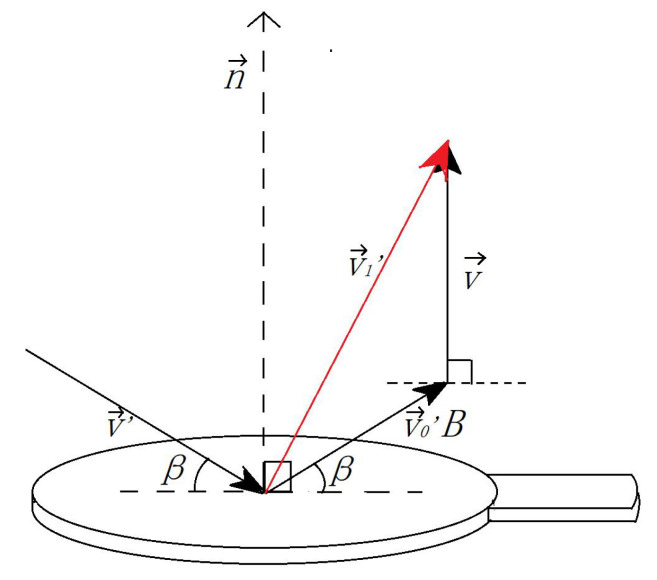
Relationship between rebound speed and speed of objects.

**Figure 12 sensors-20-04857-f012:**
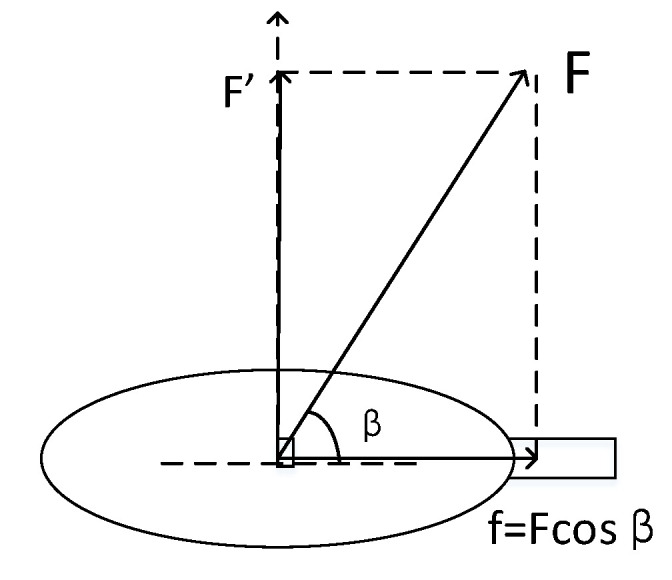
Creation of friction.

**Figure 13 sensors-20-04857-f013:**
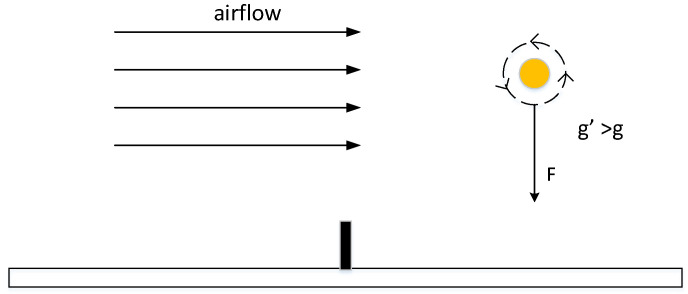
The pressure difference between the velocity of the ball up and down creates a downward force on the ball.

**Figure 14 sensors-20-04857-f014:**
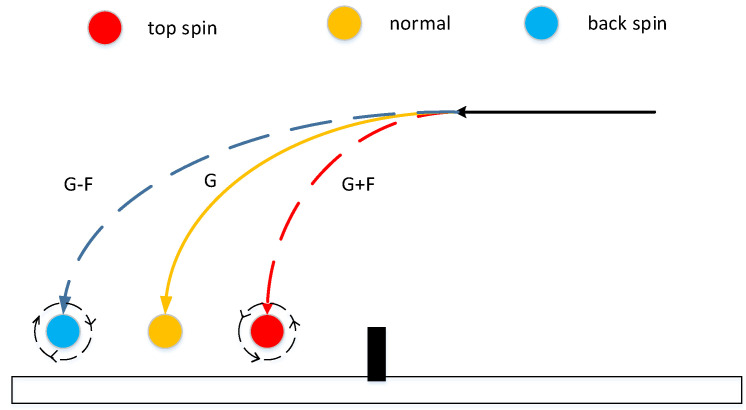
The difference between ball with top spin and ball with back spin.

**Figure 15 sensors-20-04857-f015:**
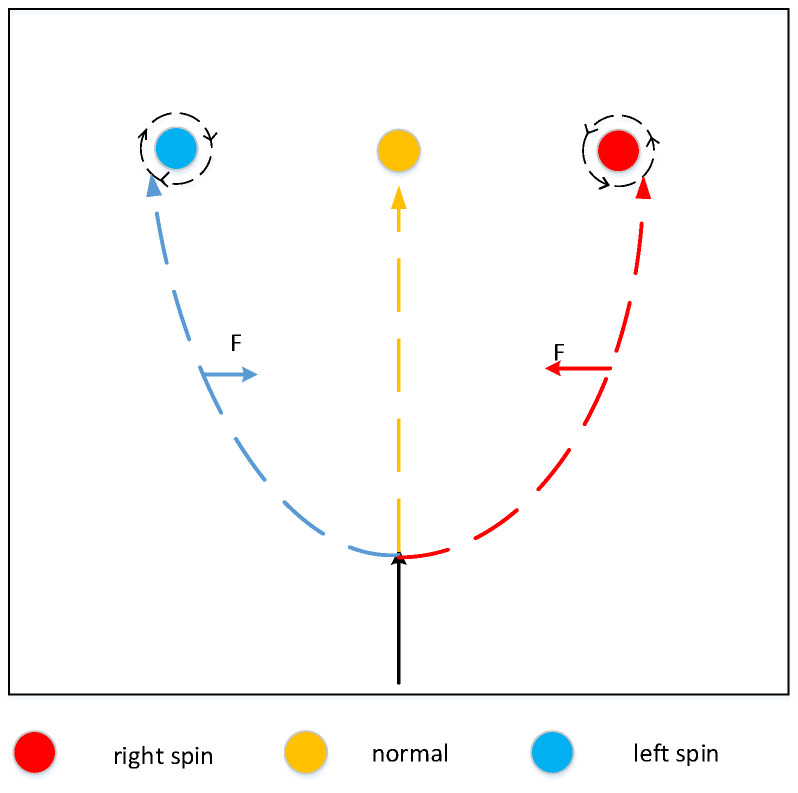
The difference in the placement of the ball with side spin.

**Figure 16 sensors-20-04857-f016:**
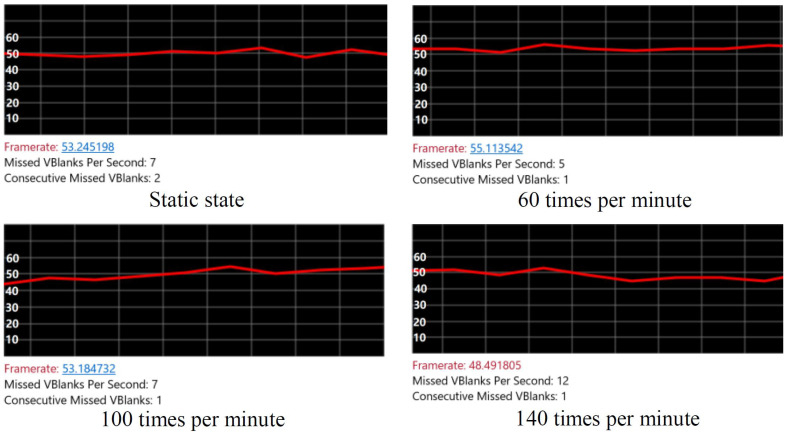
Frames per second (FPS) test results.

**Figure 17 sensors-20-04857-f017:**
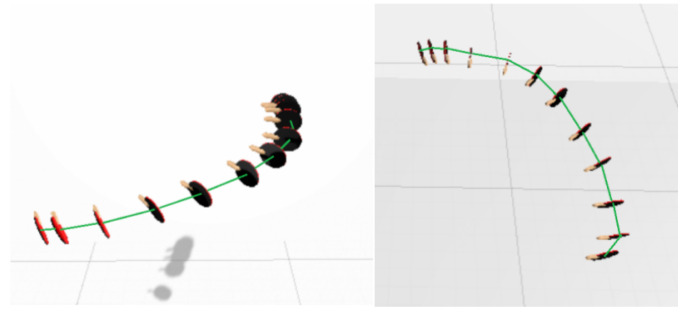
(**left**) fast swing trajectory simulation; (**right**) slow swing trajectory simulation.

**Figure 18 sensors-20-04857-f018:**
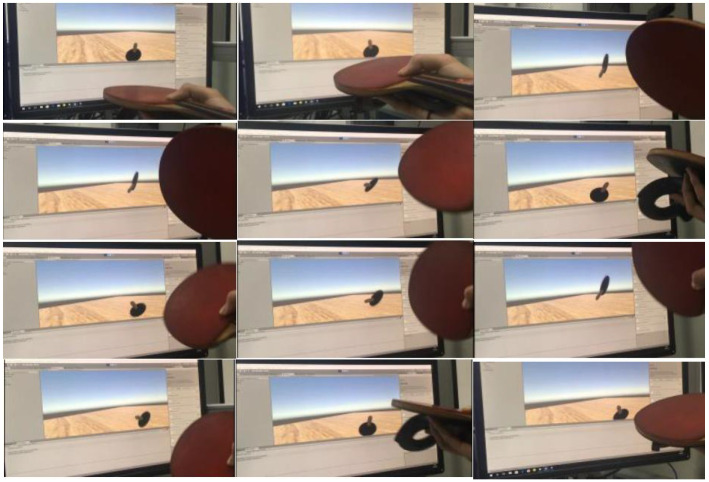
All of figures show experimental results of interactive realtime.

**Figure 19 sensors-20-04857-f019:**
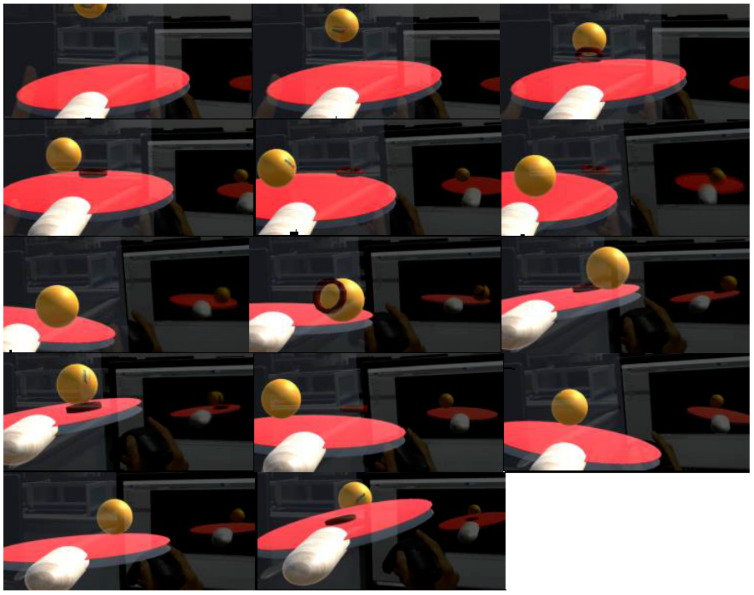
All of figures show simulated playing table tennis experiment.

**Figure 20 sensors-20-04857-f020:**
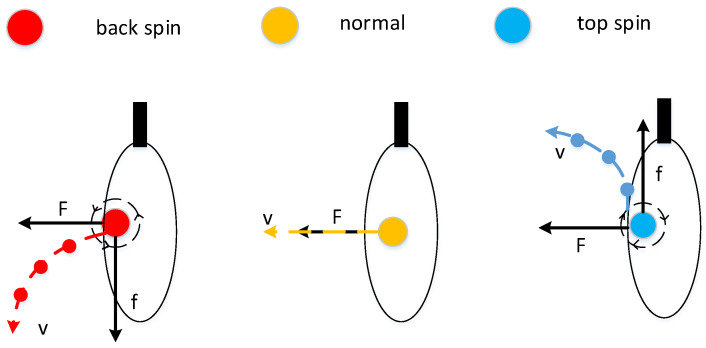
The track of the spinning ball after hitting the racket.

**Table 1 sensors-20-04857-t001:** Comparison of Motion Capture or Space Locating Technology.

	Representative	Anti-Occlusion	Accuracy	Anti-Interference	Low Cost	Real-Time	Other
Laser locating	HTC Vive-Lighthouse	not good	good	good	not good	good	poor durability
Infrared optical locating	Oculus Rift	not good	good	good	not good	good	Small coverage
Visible light locating	PS VR	not good	not good	not good	good	good	Small coverage
Ultrasonic locating	Hexamite HX11	not good	good	not good	good	not good	
Computer vision motion capture	Leap Motion	not good	not good	not good	good	good	Large amount of calculation
Based on inertial sensors	Perception Neuron	good	good	good	good	good	Accumulation error exists

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
