# Peer review of "Virtual and Real-Time Synchronous Interaction for Playing Table Tennis with Holograms in Mixed Reality"

_sensors, 2020, doi:10.3390/s20174857_

Round 1

Reviewer 1 Report

The authors present an interesting algorithm to improve interaction in a mixed reality environment. The work is worth being published provided that significant changes are introduced in the way it is presented.

  1. English language is quite correct but the style shall be improved in order to meet the requirements of a scientific publication. I suggest to replace "story telling" of how the work was developed with a more formal description of the work done.
  2. I suggest to highlight the potential of this research in other applications but the game industry. To do that, the introduction shall be improved including current promising industrial applications for mixed reality in the field of: medicine (MR surgery), aerospace (augmented reality for air traffic control,  AR aircraft maintenance), production engineering, etc. 
  3. References should be numbered according to the order they appear in the text. Reference number 1 [1] is the first one cited in the text and so on.
  4. Please specify how you conduct the assessment of motion capture features for existing devices reported in Table 1.
  5. Check figures’ captions accurately in order to make them more self-explanatory. The content of the figure should be understood by the reader also without reading the full paper.
  6. Reconsider sections’ titles. They should be more similar to the standard titles that appear in experimental research papers. 1. Introduction 2. Interaction in Mixed Reality 3. Methods 4. Results 5. Conclusions
  7. In the Methods section (see previous comment), please introduce the experiment you conducted specifying the experimental set-up, the number of subjects/participants, tools and metrics used to validate the algorithm.
  8. Reconsider how to display results, e.g. try to summarize quantitative results in a table and then discuss them in the conclusions.
  9. Rewrite conclusions by clearly stating: what the paper results show, how those results are in line with the target, the impact of this research on MR industrial applications (not only game industry), further development.

Author Response

The authors present an interesting algorithm to improve interaction in a mixed reality environment. The work is worth being published provided that significant changes are introduced in the way it is presented.

1、English language is quite correct but the style shall be improved in order to meet the requirements of a scientific publication. I suggest to replace "story telling" of how the work was developed with a more formal description of the work done.

Response 1:  We have improved the style of language to meet the requirements of a scientific publication.

2、I suggest to highlight the potential of this research in other applications but the game industry. To do that, the introduction shall be improved including current promising industrial applications for mixed reality in the field of: medicine (MR surgery), aerospace (augmented reality for air traffic control,  AR aircraft maintenance), production engineering, etc. 

Response 2:  Due to the limitation of frame rate of HTC Vive and Hololens products, it is difficult to meet the requirements of some scenes with high real-time requirements. Therefore, our proposed algorithm can be applied to other scenarios that require real-time user experience in addition to games.

We have added notes and references to other areas in the fifth paragraph of the introduction.

[1]Iulia-Cristina Stanica, FloricaMoldoveanu, Maria-IulianaDascalu, Alin Moldoveanu, Giovanni-Paul Portelli, Constanta-Nicoleta Bodea:VR System for Neurorehabilitation: Where Technology Meets Medicine for Empowering Patients and Therapists in the Rehabilitation Process. ECBS 2019: 5:1-5:7

[2] Rebecca Parkes, Neil Forrest, Sarah Baillie:A Mixed Reality Simulator for Feline Abdominal Palpation Training in Veterinary Medicine. MMVR 2009: 244-246

[3]H. Regenbrecht, G. Baratoff and W. Wilke, "Augmented reality projects in the automotive and aerospace industries," in IEEE Computer Graphics and Applications, vol. 25, no. 6, pp. 48-56, Nov.-Dec. 2005, doi: 10.1109/MCG.2005.124.

[4]Gardner M, O'Driscoll L. MiRTLE (Mixed-Reality Teaching and Learning Environment): from prototype to production and implementation[J]. 2011.

3、References should be numbered according to the order they appear in the text. Reference number 1 [1] is the first one cited in the text and so on.

Response 3: We have revised the citation order of the papers.

4、Please specify how you conduct the assessment of motion capture features for existing devices reported in Table 1.

Response 4: Official description of each product. Analyze specific performance according to official indicators. The citation has been added to second paragraph of 2.1. Selection of input devices.

Links:

[1]HTC.2015.HTC Vive.Retrieved Nov 5,2018 fromhttps://www.vive.com/cn/product/

[2]https://www.oculus.com/rift-s/

[3]https://www.playstation.com/en-us/explore/playstation-vr/

5、Check figures’ captions accurately in order to make them more self-explanatory. The content of the figure should be understood by the reader also without reading the full paper.

Response 5: We have modified the contents of figure 14,15.

Fig.1, the relationship between users anddevices is established.

Fig.2, the position information get from the locator happensto be missed by the ball in 5 consecutive frames.

Fig.3: a. The racket state in interactive process. b. Thesampling points distributionmodel.

Fig.4: Real trajectory and simulated trajectory of fastswing.

Fig.5: The collider follows the action of the user inorder to complete the contact interaction.

Fig.6: The relationship between symbols.

Fig.7: The size of the racket’s collider at differentvelocity.

Fig.8.9.10.11:The collision of ball and racket is introduced.

Fig.13: The pressure difference between the velocity of the ball up and down creates a downward force on the ball.

Fig.14: The difference between upper spinning ball and lower spinning ball.

6、Reconsider sections’ titles. They should be more similar to the standard titles that appear in experimental research papers. 1. Introduction 2. Interaction in Mixed Reality 3. Methods 4. Results 5. Conclusions

Response 6: Although the title is not expanded in this order, the content is expressed in this way. We modified the title.

7、In the Methods section (see previous comment), please introduce the experiment you conducted specifying the experimental set-up, the number of subjects/participants, tools and metrics used to validate the algorithm.

Response 7: In the SYSTEM IMPLEMENTATION, we introduce the experiment we conducted specifying the experimental set - up, the number of the subjects/participants, tools and metrics, informs the to validate the algorithm.

We have added the method part, which includes experimental set-up, the number of subjects/participants, tools and metrics used to validate the algorithm

8、Reconsider how to display results, e.g. try to summarize quantitative results in a table and then discuss them in the conclusions.

Response 8: The experimental results are explained in the SYSTEM IMPLEMENTATION and CONCLUSIONS, and we will revise the structure of the paper. The results section was added to the experiment to record the results of the experiment.

9、Rewrite conclusions by clearly stating: what the paper results show, how those results are in line with the target, the impact of this research on MR industrial applications (not only game industry), further development.

Response 9: We have rewritten the conclusion to add applications and prospects for the technology in other areas.

Reviewer 2 Report

The paper shows the results of a research where HTC Vive Controller and Microsoft Hololens are integrated together thanks to a C# application. The virtual simulation of the bouncing of a ball on a table tennis racket is simulated. Problems related to the modelling of trajectories both with and without ball spinning (and following forces due to physical effects) are described in the paper: a mathematical model of the collision detection, trajectory simulation and trajectory change after the impact with racket is described. The paper is interesting and the description of the methodology is clear.

The description of the trajectory model and collision detection depending on racket speed are the main strength points of the paper. On the other hand, the weakness points mainly relate: the way in which this research contribute to add new knowledge in the field and where it differentiate from current knowledge; some comments on the use of the AR and tracking system in real environments where there are different lighting conditions, shadows, reflections and other disturbances that usually aren’t considered in the “controlled” laboratory environment.  

In my point of view, the paper could be improved addressing the following issues:

  1. I suggest to include some notes on the effect of environment lighting/shadows on the tracking.
  2. Could you suggest possible directions to follow to simulate ball deformation under impact with the racket? Could you propose feasible ways to increase the simulation fidelity with fast hand movements (and following effective collision detection)? Do you think it’s only a problem of hardware computational power or do you think that software and algorithm advances could help in this?.
  3. There are several games and applications where the user can simulate tennis, golf and other sports with controllers in VR/AR ( see for example: https://www.augmentedrealitytennis.com/ ; https://hellofuture.orange.com/en/holotennis-future-sport-virtual-augmented-reality-meet/ ;  https://www.stereolabs.com/blog/stereolabs-brings-multiplayer-ar-to-vr-headsets/ ). Could you better explain where your system is different respect to available entertainment products? Could you specify in a more effective way where your paper is innovative respect to the present technical/scientific knowledge in the field?
  4. the English language must be improved because there are several errors: e.g. third person verbs without ending “s".

Author Response

1、I suggest to include some notes on the effect of environment lighting/shadows on the tracking.

Response 1: Because our paper focuses on the real-time hitting of table tennis, we ignore the factors such as light and shadow, and we will consider the influence of light in the later period, so as to bring more real experience.

We put the prospect of shadow and light technology in the last paragraph of theconclusion.

2、Could you suggest possible directions to follow to simulate ball deformation under impact with the racket? Could you propose feasible ways to increase the simulation fidelity with fast hand movements (and following effective collision detection)? Do you think it’s only a problem of hardware computational power or do you think that software and algorithm advances could help in this?.

Response 2: Traditional collision detection algorithms are difficult to detect the collision between real objects and virtual objects in high-speedmotion.Because the location information of real target needs to be updated in real time, collision detection is easy to be lost in high-speed operation.Instead of using gesture detection technology (the frame rate of Hololens itself cannot meet real-time requirements), we used HTC VIVE locator as the main input device and established data exchange channel for the device, enabling the system to update the racket motion status in real time.If rapid hand movement with simulation fidelity (and subsequent effective collision detection) is to be improved, the problem of collision loss caused by insufficient real-time computing power can be avoided by increasing the thickness of the collision body.

3、There are several games and applications where the user can simulate tennis, golf and other sports with controllers in VR/AR ( see for example: https://www.augmentedrealitytennis.com/ ; https://hellofuture.orange.com/en/holotennis-future-sport-virtual-augmented-reality-meet/ ;  https://www.stereolabs.com/blog/stereolabs-brings-multiplayer-ar-to-vr-headsets/ ). Could you better explain where your system is different respect to available entertainment products? Could you specify in a more effective way where your paper is innovative respect to the present technical/scientific knowledge in the field?

Response 3: Different on entertainment products: under the high speed motion sensor is hard to real-time capture movement trajectory, the existing entertainment products in real time is not guaranteed, it will bring a certain delay of user experience, and our approach is based on the speed of the swinging adaptive increase the thickness of the collision body to avoid the loss of collision detection, which can avoid the condition of the missing some collision detection under high-speed movement, to improve the authenticity of the simulation.It maintains good real-time interaction.

4、the English language must be improved because there are several errors: e.g. third person verbs without ending “s".

Response 4: We have fixed some grammatical errors.

Reviewer 3 Report

This paper present interaction methods and ball simulation for virtual table tennis. Also, this paper includes a detailed description of the interaction method. 

First of all, this paper is necessary to revise the contributions of this paer in comparision with exisitng papers (e.g. specillay rotation trajectory simulation).  

And, In the tilte, the word "holograms" makes a bit confusing.It is recomended to change the title with main contributions of this paper if possible.

And, In the result of experiments, this paper need additional experimental results such as collision handling or rotation trajectory including FPS.

And, this paper need a picture of the overall system schematic, In Fig.1 it is simply expressed, so it is difficult go understand the overall structure. for example, How the user experience the virtual table tennis content?, and what the user is holding? what is the virtual object or real object?

Additionally, authors may need to correct errors such as some space spacing errors. 

Author Response

1、First of all, this paper is necessary to revise the contributions of this paer in comparision with exisitng papers (e.g. specillay rotation trajectory simulation).  

Response 1: Our contribution in this paper is to solve the real-time problem brought by the equipment through the algorithm, and at the same time to consider all kinds of hitting and rotation problems of table tennis, which is more real.

We have revised our contributions to compare them with existing papers in introduction.

In the INTRODUCTION of the thesis, we mainly introduced our contribution:1. The MR device lacks a matched locator. In order tomakethe imaging device work with locator, we need toprovide useful information collected by the locator to theimaging device by the computer.2. It is hard to detect the collision between the ball andthe racket when the racket is moving fast.3. In order to realize the real physical interaction, usersshould be able to control the racket and hitting power toadjust the angle and velocity of the ball flight.These difficulties are problems existing in the existing technology, and they are also problems solved in our paper.4. It is also difficult to calculate the speed and direction of the spinning ball. According to the rotation of the ball, the different positions of the ball and the direction of movement of the ball hitting the racket should be considered.

2、And, In the tilte, the word "holograms" makes a bit confusing.It is recomended to change the title with main contributions of this paper if possible.

Response 2: We have modified some of the fonts in the text.

3、And, In the result of experiments, this paper need additional experimental results such as collision handling or rotation trajectory including FPS.

Response 3: We added the experimental analysis of the rotation trajectory of the ping-pong ball in the result of paper.

In result of paper,fig.20 shows the track of the spinning ball after hitting the racket.Because it is difficult to directly capture the rotation of the real-time situation, so diagram is used to simulate the real movement and the rotation of the different ball with the racket after collision, rotating the ball with the racket to produce a corresponding friction, and the friction and the racket rebound will work together in the ball movement, resulting in the ball produce different trajectories.

4、And, this paper need a picture of the overall system schematic, In Fig.1 it is simply expressed, so it is difficult go understand the overall structure. for example, How the user experience the virtual table tennis content?, and what the user is holding? what is the virtual object or real object?

Response 4: In Figure 1, User represents the user, while Locator obtains the gesture's motion trace, while the corresponding Racket is virtual Racket, when the received gesture is located, it is passed to Workstation, then Hololens obtains the corresponding data information and presents it to the user.Figure 1 will be described more accurately later.

This passage describes Figure 1 in SYSTEM DESIGN of the paper:

As shown in Fig.1, the relationship between users anddevices is established. First, we use a workstation to runtwo Unity3d projects simultaneously, supporting two setsof peripherals. One project connects HTC Vive Controller,which receives the Transform information collected by thelocator, and the other project connects Microsoft Hololensand calculates the results of collision detection and rebound.A text document can be used to transfer Position and Rotationparameters in real time between the two projects. Unity3d enginecan read and write parameters in real time through editedC# script.Next, we use Holographic Remoting Player [9], aMicrosoft Hololens suite to render Unity3d images in realtime on Microsoft Hololens.Just use Wi-Fi to connect HolographicRemoting Player and Microsoft Hololens. MicrosoftHololens can also send part of the input information to theworkstation, such as user movement, view change, terrain ofreal environment and user gestures, voice input.Finally, themixed reality image is passed to Microsoft Hololens.

5、Additionally, authors may need to correct errors such as some space spacingerrors. 

Response 5: We have fixed some measurement errors in the paper.

Round 2

Reviewer 2 Report

The paper has been overall improved. However, the English language should be revised once more in order to fix grammar errors.

Author Response

1. The paper has been overall improved. However, the English language should be revised once more in order to fix grammar errors.

Our response: Thanks the reviewer's comments. We have fixed some grammatical errors.

Reviewer 3 Report

First of all, I still think I need a picture of the overall system architecture.

For example, Is the user real? Is the ball virtual? Is the racket real? 

I would like to add a picture of a situation where a user is holding a racket and an imaginary ball is flying.

And, This paper still needs to compare previous research papers. (precise collision detection with virtual objects, rotation trajectory, etc.)

If authos have revision the paper, with a useful topic, it will be a helpful topic for our readers.

.

Author Response

1) First of all, I still think I need a picture of the overall system architecture. For example, Is the user real? Is the ball virtual? Is the racket real?  I would like to add a picture of a situation where a user is holding a racket and an imaginary ball is flying.

Our response: According to the reviewer‘s comments, we revised the figure.1 to illustrate the above problems.

2) And, This paper still needs to compare previous research papers. (precise collision detection with virtual objects, rotation trajectory, etc.) If authos have revision the paper, with a useful topic, it will be a helpful topic for our readers.

Our response:We have added comparisons in the introduction. As follows:

VR System for Neurorehabilitation[14] only considers virtual technology, but does not combine real technology, so the real experience of the scene is poor. Meanwhile, because it is mainly targeted at Neurorehabilitation experiment, and does not consider the real-time problem in the experiment.MR For Feline Abdominal Palpation Training is an early application of mixed reality technology. Due to the limitation of computing ability of early devices, many necessary experimental operations are ignored in [15].[16] and [17] mainly emphasize the application of Augmented reality in the field of automobile and aerospaceindustries. It mainly involves some basic experiences, but does not involve high-real-time operations, such as parts repair and virtual driving. At present, most of the papers have avoided the real-time problem, because it is limited by the hardware conditions, and the real-time ignored by many systems will inevitably lead to poor experience (reduced authenticity). However, we solve the device delay problem through the corresponding algorithm, to bring better experience for users.